# Differential Signaling Pathways Identified in Aqueous Humor, Anterior Capsule, and Crystalline Lens of Age-Related, Diabetic, and Post-Vitrectomy Cataract

**DOI:** 10.3390/proteomes13010007

**Published:** 2025-02-03

**Authors:** Christina Karakosta, Martina Samiotaki, Anastasios Bisoukis, Konstantinos I. Bougioukas, George Panayotou, Dimitrios Papaconstantinou, Marilita M. Moschos

**Affiliations:** 1School of Medicine, National and Kapodistrian University of Athens, 11527 Athens, Greece; 2Biomedical Sciences Research Center “Alexander Fleming”, 16672 Vari, Greece; samiotaki@fleming.gr (M.S.); panayotou@fleming.gr (G.P.); 3Department of Ophthalmology, Bristol Eye Hospital, Bristol BS1 2LX, UK; mpisanasiatr@gmail.com; 4School of Medicine, Faculty of Health Sciences, Aristotle University of Thessaloniki Greece, 54124 Thessaloniki, Greece; mpougioukas@auth.gr; 51st University Eye Clinic of Athens, “G. Gennimatas” General Hospital of Athens, National and Kapodistrian University of Athens, 11527 Athens, Greece; dpapaconstantinou@hotmail.com; 6Department of Electrophysiology of Vision, 1st University Eye Clinic of Athens, 11527 Athens, Greece; moschosmarilita@yahoo.fr

**Keywords:** aqueous humor, anterior capsule, cataract, lens, age, diabetes, vitrectomy, proteomics

## Abstract

**Background**: The purpose of this study was to detect proteomic alterations and corresponding signaling pathways involved in the formation of age-related cataract (ARC), diabetic cataract (DC), and post-vitrectomy cataract (PVC). **Methods**: Three sample types, the aqueous humor (AH), the anterior capsule (AC), and the content of the phaco cassette, were collected during phacoemulsification surgery. The samples were obtained from 12 participants without diabetes mellitus (DM), 11 participants with DM, and 7 participants without DM, with a history of vitrectomy surgery in the past 12 months. The Sp3 protocol (Single-Pot, Solid-Phase, Sample-Preparation) was used for the sample preparation. The recognition and quantification of proteins were carried out with liquid chromatography online with tandem mass spectrometry. The DIA-NN software was applied for the identification and quantification of peptides/proteins. Statistical analysis and data visualization were conducted on Perseus software. Data are available via ProteomeXchange. **Results**: A very rich atlas of the lens and AH proteome has been generated. Glycosaminoglycan biosynthesis and the non-canonical Wnt receptor signaling pathway were differentially expressed in ARC compared to both the DC and PVC groups. In the PVC group, complement activation was differentially expressed in AH samples, while glutathione metabolism and oxidoreductase activity were differentially expressed in AC samples. Microfilament motor activity, microtubule cytoskeleton organization, and microtubule binding were differentially expressed in the DC and PVC groups in both AH and AC samples. **Conclusions**: The results of this study expand the existing knowledge on pathways involved in the pathophysiology of cataract, and suggest possible important druggable targets for slower progression or even prevention of cataract.

## 1. Introduction

Cataract is the loss of normal transparency of the ocular lens, and it is the leading cause of blindness worldwide, being responsible for 33% of partial visual impairment and for 51% of severe impairment [1,2,3]. Aging is the major risk factor for cataract formation, followed by smoking, diabetes mellitus (DM), exposure to ultraviolet light, and ocular trauma, including intraocular surgery [3,4,5]. Based on the anatomic area of the opacity and LOCS III (Lens Opacities Classification System III), cataract is sub-grouped as nuclear, cortical, posterior subcapsular, and anterior subcapsular (rare) [6,7]. The physiology of cataract involves the insolubility of crystalline proteins and the formation of high-molecular-weight aggregates [8,9]. Currently, the only option for cataract treatment remains surgical intervention. The comprehensive investigation of cataract formation is essential in order to identify non-invasive therapeutic targets. Proteomic analysis provides a useful tool to investigate pathological mechanisms underlying disease. In 2023, Cantrell et al. analyzed protein fractions of 16 human lenses, applying data-independent acquisition (DIA) mass spectrometry, in order to characterize age-related changes, particularly concerning the lens microcirculation system and oxidative stress response. The results showed that the accumulation of certain age-related changes in proteome remodeling leads to reduced fiber cell permeability, which in turn causes the formation of age-related nuclear cataracts [10].

The purpose of the present study is to identify proteomic changes and corresponding signaling pathways involved in the formation of diabetic cataract (DC), age-related cataract (ARC), and post-vitrectomy cataract (PVC), using DIA mass spectrometry proteomics.

## 2. Materials and Methods

### 2.1. Human Subjects

This study was conducted according to the Declaration of Helsinki and the ethical principles of medical research involving human subjects. Informed signed consents were obtained from all patients. The clinical data were analyzed and anonymized for patients’ confidentiality. Ethical approval (18534/20 July 2022) was granted by the institutional ethics board of the hospital.

All patients diagnosed with cataract, who were scheduled to have cataract surgery at the Ophthalmology Department, were eligible for this study. Three patient groups were created. Group 1 (diabetic cataract—DC) included participants with DM (type 1 or 2) who were younger than 65 years; group 2 (age-related cataract—ARC) included participants without DM who were older than 75 years; and group 3 (post-vitrectomy cataract—PVC) included participants of any age without a history of DM, who had vitrectomy surgery for retinal detachment with gas tamponade in the last 12 months. Inclusion criteria for all 3 groups were cataract present during slit lamp examination and visual acuity lower than 20/40.

Exclusion criteria for all groups were history of ophthalmic trauma and chronic use of steroids, topical or systemic. For every patient included in this study, a detailed medical history was recorded. In, iris color, type of cataract, habits (alcohol or smoking), sun exposure time and the use of sunglasses, other medical problems (hypertension, thyroid disease, glaucoma, age-related macular degeneration), diet supplementary intake, and previous use of estrogens were recorded.

### 2.2. Sample Types

Three sample types were obtained from each participant, i.e., aqueous humor, anterior capsule, and the content of the phacoemulsification (phaco) cassette. The composition of aqueous humor is similar to blood, as proteins and metabolites are filtered from blood to aqueous humor and may reflect systemic conditions [11]. Aqueous humor is largely comprised of electrolytes, organic solutes, growth factors, cytokines, and proteins [12]. The anterior capsule samples are mainly comprised of lens epithelial cells and changes in protein expression were investigated in the present study. The phaco cassette samples are principally comprised of lens fiber cells. The majority of lens fiber cells are fully differentiated and organelles are lost. Thus, any changes in protein levels may be interpreted as changes in protein modification/degradation. Differences in AH samples could reflect systemic differences (e.g., in diabetes).

### 2.3. Sample Collection

All three sample types were collected from each patient. All cataract surgeries were performed by the same experienced surgeon to minimize operation time. During phaco surgery, a clear corneal incision was made using an I-Knife keratome and 0.2 mL of aqueous humor was collected from the anterior chamber using an insulin syringe attached to a 30-gauge needle. The anterior capsule was collected using Utrata forceps, once capsulorhexis was completed, through the main incision, and was stored in a sterile plastic box, which was filled with 2 mL of Balanced Salt Solution (BSS). The time of the surgery was limited to less than 5 min and the operation and sample collection was performed by the same experienced surgeon. At the end of the surgery, the content of the phaco cassette bag was collected from the phaco machine (Centurion^®^ Vision System, Alcon, Geneva, Switzerland). The phaco cassette contained BSS with the phacoemulsified pieces of crystalline lens along with the re-secreted aqueous humor, taking into account that the usual rate of aqueous humor production is 2.5 µL/min and the range of operation time was 5–10 min (Figure 1). All samples were stored in a −80 °C freezer (Haier Biomedical, Qingdao, China). The samples were transferred to the lab on dry ice.

### 2.4. Sample Preparation

The anterior capsule samples were lysed and homogenized in a buffer, which consisted of 4% SDS and 0.1 M DTT in 0.1 M TEAB, and incubated for 10 min in 99 °C. Then, the samples were sonicated in a water bath for 10 min, and this procedure (heating and sonication) was repeated twice. The following step of sample preparation was common for the three sample types.

In each phaco cassette content sample, an equal amount of ethanol 100% was added together with 40 μL of beads (1:1 mixture of hydrophilic and hydrophobic SeraMag carboxylate-modified beads (Cytiva, Marlborough, MA, USA) for each 250 mL of sample). Speedbead magnetic carboxylate-modified particles offer a rapid magnetic response and high binding capacity, along with a large surface area, excellent sensitivity, stability, physical durability, and fast reaction kinetics. The samples were mildly agitated for 30 min in order to let the proteins be captured onto the beads. The samples were subjected to centrifugation at 2200 REV/Min for 10 min in order to collect the magnetic beads and the supernatant was removed. Centrifugation was repeated. The next step of sample preparation was common for all three sample types.

The protein extracts from all three sample types were subjected to tryptic digestion using the Sp3 protocol, which included a reduction step with 100 mM DTT and alkylation with 200 mM iodoacetamide (Acros Organics, Thermo Fisher Scientific, Waltham, MA, USA). Each sample received 20 µg of beads (a 1:1 mixture of hydrophilic and hydrophobic SeraMag carboxylate-modified beads, Cytiva, Marlborough, MA, USA) in a final concentration of 50% ethanol. Protein clean-up was conducted using a magnetic rack, where the beads were washed twice with 80% ethanol and once with 100% acetonitrile (Fisher Chemical, Thermo Fisher Scientific, Waltham, MA, USA). The proteins bound to the beads were then digested overnight at 37 °C under vigorous shaking (1200 rpm, Eppendorf Thermomixer, Hamburg, Germany) with 0.5 µg Trypsin/LysC (MS grade, Promega, Madison, WI, USA) in 25 mM Ammonium bicarbonate. The next day, peptides were purified using a modified Sp3 clean-up protocol, solubilized in mobile phase A (0.1% Formic acid in water), sonicated, and then the peptide concentration was measured by absorbance at 280 nm using a nanodrop instrument. The samples were analyzed via liquid chromatography–tandem mass spectrometry (LC-MS/MS) on a Dionex Ultimate 3000 nanoRSLC system coupled to a Thermo Q Exactive HF-X Orbitrap mass spectrometer (Thermo Fisher Scientific, Waltham, MA, USA). Peptide samples were injected directly and separated on a 25 cm C18 analytical column (PepSep, 1.9 µm beads, 75 µm ID) over a one-hour gradient, starting at 7% Buffer B (0.1% Formic acid in 80% Acetonitrile) and ramping to 35% over 40 min, followed by an increase to 45% in 5 min, a further increase to 99% in 0.5 min, and then held constant for 14.5 min for equilibration. Full MS was acquired in profile mode using a Q Exactive HF-X Hybrid Quadrupole-Orbitrap mass spectrometer, scanning from 375 to 1400 *m*/*z* with a resolving power of 120 K, an Automatic Gain Control (AGC) target of 3 × 10^6^, and a maximum injection time of 60 ms. Data-independent acquisition was performed with 8 Th windows (39 total loop counts), each using 15 K resolving power, an AGC target of 3 × 10^5^, a maximum injection time of 22 ms, and a normalized collision energy (NCE) of 26. Each sample was analyzed in three technical replicates.

### 2.5. Data Processing Protocol

Orbitrap raw data were analyzed using DIA-NN 1.8 (Data-Independent Acquisition by Neural Networks) by searching against the Human_Reviewed Proteome (downloaded from Uniprot, 50,516 protein entries, on 11 April 2022) in library-free mode. The analysis allowed for up to two missed tryptic cleavages and a maximum of three variable modifications per peptide. Neural networks were employed for peak selection. A spectral library was generated from the DIA runs and used to reanalyze the data in double search mode.

The DIA-NN search included variable modifications such as methionine oxidation, N-terminal methionine excision, and protein N-termini acetylation, with carbamidomethylation of cysteine residues set as a fixed modification. The “match between runs” feature was applied for all analyses, and the output (precursor) was filtered at a 0.01 false discovery rate (FDR). Protein inference was performed at the gene level, using only proteotypic peptides.

### 2.6. Statistical Analysis

The software Perseus (Version 1.6.15.0) was used for statistical analysis (defined groups *t*-test) and data visualization. Raw intensity values were log2-transformed, classified according to clinical groups, and then filtered based on a 50% valid value threshold. Missing values were imputed using a normal distribution.

Two-sample *t*-tests were conducted to compare the ARC-DC and ARC-PVC groups. Volcano plots were generated to visualize the *t*-test results, with the permutation-based false discovery rate (FDR) set at 0.05 and S0 (artificial within-group variance) set at 0.1. Additionally, ANOVA tests were performed for comparisons across all groups, with the results visualized in heatmaps. For the ANOVA tests, the permutation-based FDR was also set at 0.05, with S0 set at 0.1.

For gene enrichment analysis, ShinyGO (Version 0.77, Human species, permutation-based FDR cutoff 0.01, KEGG pathway database, chart-type barplot) and Metascape were used [13]. For the creation of Venn diagrams, Venny (Version 2.1.0) was used, and all the results were saved in excel files [14].

The basic characteristics (demographic data) of the patients were summarized using means and standard deviations (SDs) for continuous variables with normal distributions, and medians with interquartile ranges (IQRs) for skewed data. A two-way ANOVA (Analysis of Variance) test was conducted to assess the impact of categorical demographic factors on the values of significant proteins across the three cataract groups. An ANCOVA test was performed to evaluate the effect of the qualitative demographic factors on the values of the significant proteins for the three cataract groups. A two-sided *p*-value of less than 0.05 was considered statistically significant. All analyses of demographic data were carried out using the R programming language and RStudio IDE.

## 3. Results

### 3.1. Sample Characteristics

Group 1 included 11 patients, group 2 included 12 patients, and group 3 included 7 patients. The three sample types were collected from all patients. The demographic data of the patients included in the study are presented in Table 1. None of the patients in any group had a history of trauma, ocular inflammation, previous topical or systemic use of steroids, or estrogen intake.

### 3.2. Protein Atlas

In total, 1639 proteins were identified in aqueous humor samples, 2815 proteins in anterior capsule samples, and 2975 proteins in phaco cassette content samples. Among all three sample types, 657 proteins were common and those represent the core proteome of this study. Thirty proteins were common between aqueous humor and phaco cassette content samples. A full list of the identified proteins is provided in Appendix A. A full list of the significant identified proteins is provided in Appendix A. The Venn diagram of all identified proteins in the three sample types is shown in Figure 2 [14]. In the Human Protein Atlas, 29 genes have been analyzed in eye [15]. Comparing the combined results of this study with those of the Human Protein Atlas, 12 proteins (CRYAA, CRYAB, CRYBA1, CRYBA4, CRYBB1, CRYBB3, CRYGC, CRYGD, CRYGS, LUM, OPTC, and TGFBI) were common with the eye proteome. Figure 2 shows the Venn diagrams of the three sample types of this study and the eye proteome of the Human Protein Atlas.

Eight proteins were exclusively present in the eye proteome (BTBD6, CRYAA2, CRYGA, GJA3, GJA8, LGSN, MIP, and PRX). However, either family members or other forms of these proteins were present in our samples. In contrast, in all samples of this study combined, 4299 unique genes were identified, extending the existing dataset of the human eye.

### 3.3. Gene Enrichment Analysis

#### 3.3.1. Aqueous Humor Samples

In the dotplot for DC-ARC and ARC-PVC comparisons, proteins with differential abundance were analyzed in ShinyGO 0.77 (Figure 3a). Volcano plots of the proteins involved in the significantly different pathways between the sample groups were created (Figure 4a).

Pathways significantly impacted in ARC compared with DC included the glycosaminoglycan biosynthetic and catabolic process, cell adhesion molecules (CAMs), and ECM–receptor interaction.

Pathways significantly impacted in DC compared with ARC included the proteasome core complex, intermediate filament organization, actin–myosin filament sliding, actomyosin structure organization, lens fiber cell development, microtubule cytoskeleton organization, microtubule binding, and the tumor necrosis factor-mediated signaling pathway.

Pathways significantly impacted in ARC compared with PVC included lysosome, the sphingolipid metabolic process, the proteoglycan metabolic process, cell adhesion molecules (CAMs), oxidation, oxidoreductase activity, and lens fiber cell development.

Pathways significantly impacted in PVC with ARC included regulation of fibrinolysis and complement and coagulation cascades.

#### 3.3.2. Anterior Capsule Samples

In the volcano plot for DC-ARC and ARC-PVC comparisons, proteins with differential abundance were analyzed in ShinyGO 0.77 (Figure 3b). Volcano plots of the proteins involved in the significantly different pathways between the sample groups were created (Figure 4b).

Pathways significantly impacted in ARC compared with DC included the non-canonical Wnt receptor signaling pathway, wnt-protein binding, glycosphingolipid biosynthesis, glycosaminoglycan biosynthesis (chondroitin sulfate, heparan sulfate, and keratan sulfate) and degradation, lysosome organization, cytokine–cytokine receptor interaction, cytokine binding, negative regulation of chemotaxis, and the glycosylceramide metabolic process.

Pathways significantly impacted in DC compared with ARC included structural constituent of ribosome, nuclear mRNA splicing via spliceosome, carbon–oxygen lyase activity, oxidative phosphorylation, sensory perception of touch and pain, cytokinesis, glycolysis/gluconeogenesis, actomyosin structure organization, structural constituent of eye lens, microtubule cytoskeleton organization, and microfilament motor activity.

Pathways significantly impacted in ARC compared with PVC included the non-canonical Wnt receptor signaling pathway, the glycosaminoglycan biosynthetic process, the glycosphingolipid metabolic process, positive regulation of pathway-restricted SMAD, and cytokine–cytokine receptor interaction.

Pathways significantly impacted in PVC compared with ARC included complement activation (classical pathway), nuclear mRNA splicing via spliceosome, ribosome, cellular response to hydrogen peroxide, cortical actin cytoskeleton organization, structural constituent of eye lens, and microtubule cytoskeleton organization.

#### 3.3.3. Phaco Cassette Content Samples

In the volcano plot for DC-ARC and ARC-PVC comparisons, proteins with differential abundance were analyzed in ShinyGO 0.77 (Figure 3c). Volcano plots of the proteins involved in the significantly different pathways between the sample groups were created (Figure 4c).

Pathways significantly impacted in ARC compared with DC included the non-canonical Wnt receptor signaling pathway, glycosphingolipid biosynthesis, glycosaminoglycan biosynthesis and degradation, chemokine activity, and cytokine activity.

Pathways significantly impacted in DC compared with ARC included proteasome complex, ribosome, catalytic step 2 spliceosome, complement activation, cytokinesis, carbon–oxygen lyase activity, glycolysis/gluconeogenesis, Citrate cycle (TCA cycle), myosin binding, microtubule cytoskeleton organization, organelle transport along microtubule, structural constituent of eye lens, and microfilament motor activity.

Pathways significantly impacted in ARC compared with PVC included the non-canonical Wnt receptor signaling pathway and negative regulation of canonical Wnt receptor signaling pathway, glycosphingolipid biosynthesis, and glycosaminoglycan biosynthetic, catabolic, and metabolic processes.

Pathways significantly impacted in PVC compared with ARC included the proteasome complex, structural constituent of ribosome, nuclear mRNA splicing via spliceosome, carbon–oxygen lyase activity, regulation of complement activation, structural constituent of eye lens, actin cytoskeleton, and microtubule cytoskeleton organization.

Heatmaps of the proteins with the most significant difference between the cataract groups are presented in Figure 5.

Raincloud plots of VTN protein in AH samples, ST3GAL5 protein in AC samples, and FRZB and SFRP5 proteins in phaco cassette content samples are presented in Figure 6.

### 3.4. Demographic Data

Two-way ANOVA and ANCOVA tests were performed to evaluate the effect of demographic data of the patients on the most significant proteins of each group. These included VTN protein in aqueous humor samples, ST3GAL5 protein in anterior capsule samples, and FRZB and SFRP5 proteins in phaco cassette content samples. The tests revealed no statistically significant effect (*p* > 0.05) on protein values of the three groups for the following factors: sex, height, weight, axial length, K1, K2, smoking, alcohol consumption, exposure to the sun, use of sunglasses, hypertension, glaucoma, age-related macular degeneration, aspirin intake, use of diet supplements, and thyroid disease. It is important to note that this analysis was based on subgroup data, with relatively small sample sizes in each group. Consequently, drawing definitive conclusions may not be appropriate due to the limitations in sample size.

## 4. Discussion

### 4.1. WNT Pathway

Myosin II interacts with actin and forms the actomyosin contractile network. Actomyosin participates in lens fiber elongation, differentiation, and migration, and disruptions of its function affect lens biomechanical properties [16,17,18]. In the DC and PVC groups, myosin binding, actin–myosin filament sliding, and actomyosin structure organization pathways were significantly differentially expressed compared to the ARC group. Moreover, certain wnt proteins implicated in the canonical Wnt receptor signaling pathway (WNT11, WNT2B, WNT4, WNT5A, WNT5B, WNT7A, and WNT7B) were significantly differentially expressed in anterior capsule samples of the ARC group, while other myosin proteins implicated in the same pathway (MYH6, MYH7, MYH9, MYH10, MYH11, and MYH14) were significantly differentially expressed in the DC group. Myosin overexpression in the DC group may suggest a possible role in the formation of diabetic cataract, through disruption in the arrangement of lens fiber cells. TGFβ was differentially expressed in anterior capsule samples of the ARC group. The non-canonical Wnt receptor signaling pathway and the negative regulation of the canonical Wnt receptor signaling pathway were both significantly differentially expressed in the ARC group. Wnt/planar cell polarity pathways play a crucial role in actin filament cytoskeletal organization during fiber cell differentiation [19,20]. The canonical Wnt (or Wnt/β-catenin) pathway consists of four components: the extracellular signal, the membrane segment, the cytoplasmic segment, and the nuclear segment. Extracellular signals are mainly provided by Wnt proteins such as Wnt3a, Wnt1, and Wnt5a [21,22].

The cell membrane segment mainly contains the Wnt receptors Frizzled (specific seven-pass transmembrane receptor Frizzled protein) and LRP5/6 (single-pass lipoprotein receptor-related protein) [23,24]. The cytoplasmic and nuclear segment mainly includes β-catenin. The non-canonical Wnt signaling pathway is β-catenin-independent and can inhibit the canonical pathway at multiple levels [25]. Secreted frizzled-related proteins (SFRPs) consist of five secreted glycoproteins, which act as negative modulators of the Wnt signaling pathway, and particularly SFRP3 (or FRZB) and SFRP5 are shown to act as antagonists for the canonical Wnt signaling pathway [26]. In our study, both FRZB and SFRP5 proteins were significantly differentially expressed in ARC samples compared to both DC and PVC (*p* < 0.0001) and may participate in cataract formation through regulation of the canonical Wnt signaling pathway. The non-canonical Wnt signaling pathway participates in the regulation of fiber differentiation, migration, and elongation via reorganization of the actin cytoskeleton and microtubule network [19,27,28]. Studies have shown that Wnt5a protein is highly expressed in cataractous plaques in animal models, particularly in cells near the lens capsule. This suggests that Wnt signaling plays a role in regulating abnormal growth and differentiation processes in TGFß-induced cataracts [29]. Furthermore, recent research indicates that the canonical Wnt signaling pathway may contribute to the abnormal proliferation and migration of lens epithelial cells following low-dose irradiation [30,31].

### 4.2. Glycosaminoglycans

Glycosaminoglycans (GAGs) are a group of polysaccharides, which consists of hyaluronan, dermatan sulfate (DS), keratan sulfate (KS), chondroitin sulfate (CS), heparin, and heparan sulfate (HS) [32]. Proteoglycans are made of a core protein that binds to GAG chains [33]. Heparan sulfate proteoglycans (HSPGs) are an important component of the cell surface, and they are ubiquitously present in the pericellular environment and in the extracellular matrix [34]. HSPGs interrelate with various proteins, such as growth factors, cytokines, chemokines, collagens, integrins, laminin, and fibronectin [34]. This way, they regulate key biological pathways, including cell proliferation and development, immune response, and tissue healing [35]. The arrangement of lens cells, in particular proliferation, survival, and fiber differentiation, is regulated by ocular growth factors. A recently published study supports the idea that lens cell proliferation and fiber differentiation, mediated by fibroblast growth factor-2, requires HSPGs [36]. HSPGs may have a significant role in specific biological events involved in lens growth and homeostasis by regulating growth factors, matrix proteins, and signaling molecules [36,37]. Decorin is a PG that regulates growth factors, cell proliferation, and migration, which may be associated with posterior capsular opacification (PCO) development in lens [38]. Decorin is considered as a possible therapeutic target of various ocular diseases, including PCO, due to its anti-fibrotic, anti-inflammatory, and antioxidant effect by altering growth factor pathways [38]. Glycosaminoglycan biosynthetic, metabolic, and catabolic processes were significantly differentially expressed in ARC compared to both DC and PVC, suggesting a possible role in ARC formation and a possible therapeutic target for lens opacification by preventing cell proliferation and repairing damaged lens tissue.

### 4.3. Glycosphingolipids

Glycosphingolipids comprise essential elements of cell membranes and participate in cellular differentiation and interaction as second messengers. Ceramide is the main regulator of the sphingolipid metabolic process and necessary for lipid signaling. Increased cellular levels of ceramide are linked to apoptosis and cell death as a response to stress [39]. Concentration of ceramides in the lens increases with age, and, by increasing apoptosis in lens epithelial cells, may be implicated in the development of ARC [40,41,42]. A recent study showed that the expression levels of ceramide in the AH of patients with DM were significantly lower compared to controls, and this may underline the different pathogeneses between DC and ARC [43]. The results of the present study revealed that both glycosphingolipid biosynthesis and the glycosylceramide metabolic process were significantly differentially expressed in ARC compared to DC and PVC, suggesting a possible involvement of glycosphingolipids and ceramide in the formation of ARC. Ganglioside GM3 synthase (ST3GAL5) is a primary glycosyltransferase for the synthesis of complex gangliosides. It has been reported that the human lens accumulates gangliosides (such as GM3, GM2, and GM1) and that age-related changes in lens glycolipids may modify the cell-to-cell interaction induced by cell surface sugar chains, leading to the formation and progression of cataract [44,45]. In our study, ST3GAL5 was significantly differentially expressed in ARC compared to both the DC (*p* = 0.042) and PVC groups (*p* < 0.0001), and it may be particularly implicated in cataract formation by modifying cell membranes and cell-to-cell interactions [5].

### 4.4. Crystallins

In phaco cassette content samples, specific crystallins (CRYBA1, CRYBA2, CRYBA4, CRYBB1, CRYBB2, CRYGC, CRYGD, and CRYGS) were commonly differentially expressed in ARC compared to the DC and PVC groups. The differential expression of βγ-crystallins in ARC indicates their role in lens transparency and ARC formation. In both the anterior capsule and phaco cassette content samples, the proteins implicated in the pathway of structural constituent of the eye lens (BFSP1, BFSP2, CRYAA, CRYAB, CRYBA1/2/4, CRYBB1/2/3, CRYGC, CRYGD, CRYGS, KRTs, and VIM) and in the MIP pathway (AQP1, AQP5) were significantly differentially expressed in the ARC group. CRYAA acts as a chaperone that prevents the aggregation of βγ-crystallins, and the significant difference in its presence between the groups reinforces its role in ARC formation [46]. Various studies have shown that aquaporins contribute to the maintenance of lens transparency, and the results of the present study support these data, with the differentially expressed AQP1 and AQP5 in the anterior capsule samples of the ARC group [47,48,49].

### 4.5. Lens Cytoskeleton

Lens fiber cell maturation is assisted by three cytoskeletal systems: microtubules, intermediate filaments, and actin filaments [16]. Microtubules, arranged along the long axis of lens fibers, are essential for cell elongation and vesicular transport [50]. Beaded intermediate filaments (intermediate filament proteins, BFSP1 and BFSP2), are needed for mechanical integrity and lens transparency, while actin is necessary for fiber cell packing and mechanical stiffness [16]. Our results showed that the microtubule cytoskeleton organization, microtubule binding, microtubule organizing center, and organelle transport along microtubule pathways were significantly differentially expressed in the DC and PVC groups compared to the ARC group. Microtubules might participate in lens development and maintenance, based on the results of studies that showed loss of microtubules in cataractous lenses [51]. Previous reports in cataract models demonstrated specific changes in cytoskeletal proteins (tubulin, actin, vimentin, spectrin, and the lens-specific filaments filensin and CP49) during normal development of the transparent lens, as well as a dramatic loss of cytoskeletal protein during cataract formation [52]. In addition to this, the microfilament motor activity and intermediate microfilament pathways were commonly differentially expressed in the DC and PVC groups compared to the ARC group. BFSP1 was significantly differentially expressed in the DC and PVC groups of phaco cassette content samples. Vimentin, BFSP1 and BFSP2 contribute to the normal maintenance of the eye lens [53]. In particular, it is reported that BFSP1 and BFSP2 are essential for lens transparency, since they slow down lens aging, and that mutations in BFSP1 and BFSP2 can cause cataract [54].

### 4.6. TGF-β/SMAD Pathway

The SMAD pathway is a signaling pathway necessary for the transmission of signals from the cell surface to the nucleus, particularly in the context TGF-β signaling. The TGF-β/SMAD pathway is involved in the regulation of lens epithelial cell proliferation, differentiation, and lens fiber cell development [55]. Downregulation of HTRA1, which is a serine protease and a regulator of the TGF-β signaling pathway, can significantly upregulate *p*-Smad2/3 and activate the TGF-β/Smad signaling pathway. This may result in irregular proliferation and abnormal arrangement of lens epithelial cells, and thus in subcapsular cataract formation [56]. Our results showed significantly differentially expressed positive regulation of pathway-restricted SMAD protein phosphorylation in anterior capsule and phaco cassette content samples of the ARC group.

### 4.7. Spliceosome

The spliceosome is a large RNA–protein complex responsible for the catalysis of the removal of introns from nuclear pre-mRNA [57]. In our study pathways of the C complex spliceosome, catalytic step 2 spliceosome and positive and negative regulation of nuclear mRNA splicing via spliceosome were significantly differentially expressed in anterior capsule and phaco cassette content samples of the DC and PVC groups compared to the ARC group. A recent study showed that a splicing deletion mutation is associated with autosomal recessive congenital cataract [58]. Changes in alternative splicing may affect the production of protein isoforms and impact lens cell function and transparency.

### 4.8. Ribosome

Decreased expression of ribosomal proteins might lead to decreased lens protein synthesis, and decreased translation of proteins is linked to ARC, as shown in previous animal studies [59]. In addition to this, ultraviolet-B radiation is shown to cause alterations in protein synthesis in cultured lens epithelial cells, and these alterations may contribute to cataract formation [60]. Our results showed significant differential expression of structural constituents of the ribosome pathway in phaco cassette content and anterior capsule samples of DC and PVC. A study demonstrated that ribosomal protein L21, and to a lesser extent L15, L13a, and L7a, underwent decreased expression in human cataractous lenses compared with normal ones [61]. These ribosomal proteins, among others, were also decreased in the ARC group of our study, and this finding is likely related to aging epithelial cells in the AC sample having reduced protein synthesis.

### 4.9. Proteasome

The proteasome is a protease complex responsible for the hydrolysis of proteins, and as a result for the removal of aged or aberrant proteins [62,63]. Proteasome activity was reported to be decreased with increasing lens opacification. Consequently, the maintenance of proteasome activity, even in the oldest central lens fibers, might be essential for lens transparency [63]. Failure of proteasome activity may result in the accumulation of damaged proteins, lens opacification, and cataract [63]. Accordingly, in our study, the 20 S and 26 S proteasome pathways were significantly differentially expressed in aqueous humor samples of the ARC group compared to the DC group, while the proteasome complex pathway was significantly differentially expressed in phaco cassette content and anterior capsule samples of ARC compared to both the DC and PVC groups.

### 4.10. Oxidative Stress

Oxidation–reduction (Redox) reactions involve the transfer of electrons between molecules [64]. Redox reactions play a role in the generation of reactive oxygen species (ROS) and oxidative damage to lens proteins [65]. Oxidoreductases are enzymes that catalyze redox reactions and in the lens; they include superoxide dismutase, catalase, and glutathione peroxidase. Dysfunction or decreased activity of these enzymes can lead to increased oxidative stress and cataract formation. As a result, the balance between oxidative and antioxidant systems is a key factor in cataract formation. Holekamp et al. reported that vitrectomy surgery significantly increases intraocular oxygen tension during the procedure and for a prolonged postoperative period, and as a result the crystalline lens is exposed to abnormally high oxygen, which may lead to nuclear cataract formation [66]. In anterior capsule samples of the PVC group, the oxidative stress pathway, oxidation–reduction pathway, oxidoreductase, and glutathione were uniquely differentially expressed, underlying a possible protective role of the lens capsule epithelium cells against oxidative stress in vitrectomized eyes, which, once overcome by oxidative stress, leads to cataract.

### 4.11. Complement Pathway

Complement is a part of the immune system involved in inflammation. Cell damage stimulates complement activation and, vice versa, complement activation leads to cell damage and inflammation [67]. Vitronectin (VTN) is a glycoprotein, whose functions include promotion of cell adhesion, binding to several serine protease inhibitors (serpins), and inhibition of complement-mediated cytolysis either by blocking C5b-7 membrane binding or by preventing C9 polymerization [68]. A recent study investigated the metabolic characterization of aqueous humor in humans to assess the cataract progression after pars plana vitrectomy and demonstrated significantly changed levels of pelargonic acid, which plays an essential role in the regulation of inflammatory response, and thus may contribute to cataract progression [69]. Another study demonstrated that certain cytokines (BLC, I-309, IL-6, IL-8, IP-10, and ICAM-1) can accumulate in the anterior chambers of post-vitrectomized eyes, changing the micro-environment and facilitating cataract development [70,71]. In our study, the complement activation pathway was significantly differentially expressed in all three sample types of the PVC group (C3, C4A, C4B, C8B, C8G, C9, CD55, CFH, CFHR1, CFI, MMP19, PHB1, and VTN). In particular, VTN was one of the most significantly increased proteins in the PVC group compared to the ARC group (*p* < 0.0001) (Figure 6a). VTN seems to have immunomodulatory properties and dysregulation of inflammation could potentially contribute to cell damage and post-surgery cataract formation. Vitrectomy surgery may trigger chronic inflammation and activation of the complement system, leading to cataract formation, in combination with oxidative stress.

### 4.12. Therapeutic Implications

The results of the present study provide insight into potential pharmacotherapeutic targets. In the future, alternative treatment of age-related cataract may include solid lipid nanoparticles to increase certain proteins responsible for lens transparency, such as Vimentin, BFSP1 and BFSP2, or WNT5A, or to erase proteins involved in apoptosis and cell death as a response to stress, such as ceramide. These drugs could be used either as eyedrops or injected intracameral. In addition to this, intracameral injection of nanoparticles of oxidoreductase enzymes after vitrectomy may slow down post-vitrectomy cataract progression.

## 5. Conclusions

In this study, key pathways involved in cataract formation in age-related (ARC), diabetic (DC), and post-vitrectomy (PVC) cataracts were identified. Differential expression of the non-canonical Wnt receptor signaling pathway in ARC may promote abnormal proliferation and migration of lens epithelial cells, contributing to cataract formation. Similarly, the differential expression of glycosaminoglycan and glycosphingolipid pathways in ARC suggests potential therapeutic targets for preventing cell proliferation and repairing damaged lens tissue. Differential expression of AQP1 and AQP5 in the ARC group supports their role in maintaining lens transparency, while reduced expression of vimentin, BFSP1, and BFSP2 may contribute to lens aging. Proteomic complexity, particularly the diversity of proteoforms resulting from alternative splicing and post-translational modifications, further underscores the importance of understanding protein dynamics in cataract pathogenesis. Alterations in proteoforms may affect lens clarity, making proteomic analysis crucial for identifying new therapeutic targets. In the PVC group, the significant differential expression of oxidation–reduction pathways in anterior capsule samples suggests a protective role of lens epithelial cells against oxidative stress. However, PVC formation may occur when oxidative stress surpasses protective mechanisms.

This is the first study to use phaco cassette content samples to investigate cataract formation, opening new avenues for research. Our results expand the existing knowledge on cataract profiles, highlight novel pathways involved in cataract pathophysiology, and suggest druggable targets for slowing or preventing cataract progression. Future research should focus on identifying diagnostic biomarkers and developing a grading system for cataract severity.

## 6. Limitations

An age-matched transparent lens group would be valuable in distinguishing proteins/pathways involved in cataract formation and those that are simply due to the aging process, but a clear lens is an uncommon finding in ages over 75 years old and clear lens extraction for refractive reasons is rarely preferred by this age group and may raise ethical issues. Moreover, clear lens extraction in younger patients for refractive reasons could indeed be used as a baseline, but since this patient group would include mainly high-myopic patients, it would raise concerns regarding selection bias. Further, in this study all cataract types were combined in order to provide an overall investigation of cataract formation pathways. Using subgroups for the types of cataracts may provide precious data regarding the different mechanisms that cause each distinct cataract type. Nevertheless, in this study three different sample types were collected from each patient and this may serve as a control, taking into account that the up/downregulation of pathways/proteins appears to follow a consistent pattern across all three sample types. Finally, in this study, phaco cassette samples were collected after phacoemulsification surgery. Even though phacoemulsification may generate free radical-mediated oxidative stress, studying such samples may open new horizons regarding method development that focuses on cataract investigation.

## Figures and Tables

**Figure 1 proteomes-13-00007-f001:**
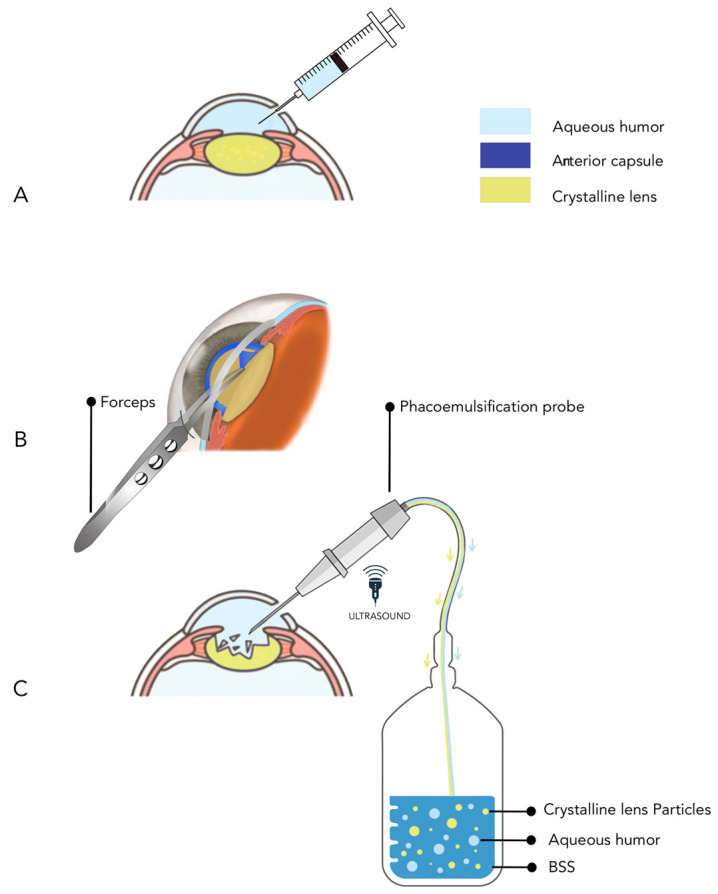
Sample collection: (**A**) The aqueous humor was collected from the anterior chamber using an insulin syringe. (**B**) The anterior capsule was collected using Utrata forceps after capsulerhexis was completed and was washed with Balanced Salt Solution (BSS) and stored immediately in a sterile box filled with BSS. (**C**) The content of the phaco cassette was collected at the end of the surgery, which contained the phacoemulsified particles of crystalline lens together with a small portion of the re-secreted aqueous humor in Balanced Salt Solution (BSS).

**Figure 2 proteomes-13-00007-f002:**
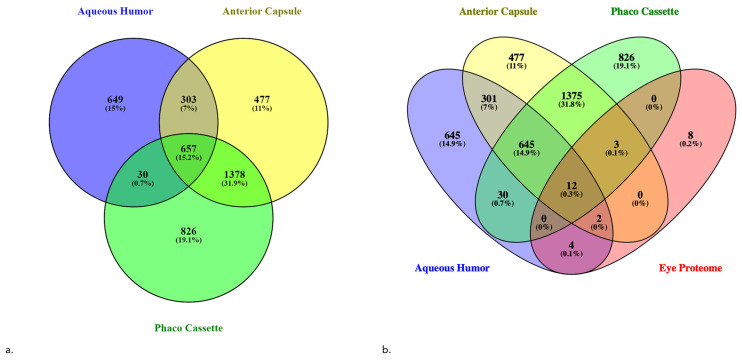
(**a**) Venn diagram of the identified proteins in aqueous humor, anterior capsule, and phaco cassette sample types. (**b**) Venn diagram of the three sample types and the eye proteome of the Human Protein Atlas, demonstrating that 12 elements (0.3%) were common.

**Figure 3 proteomes-13-00007-f003:**
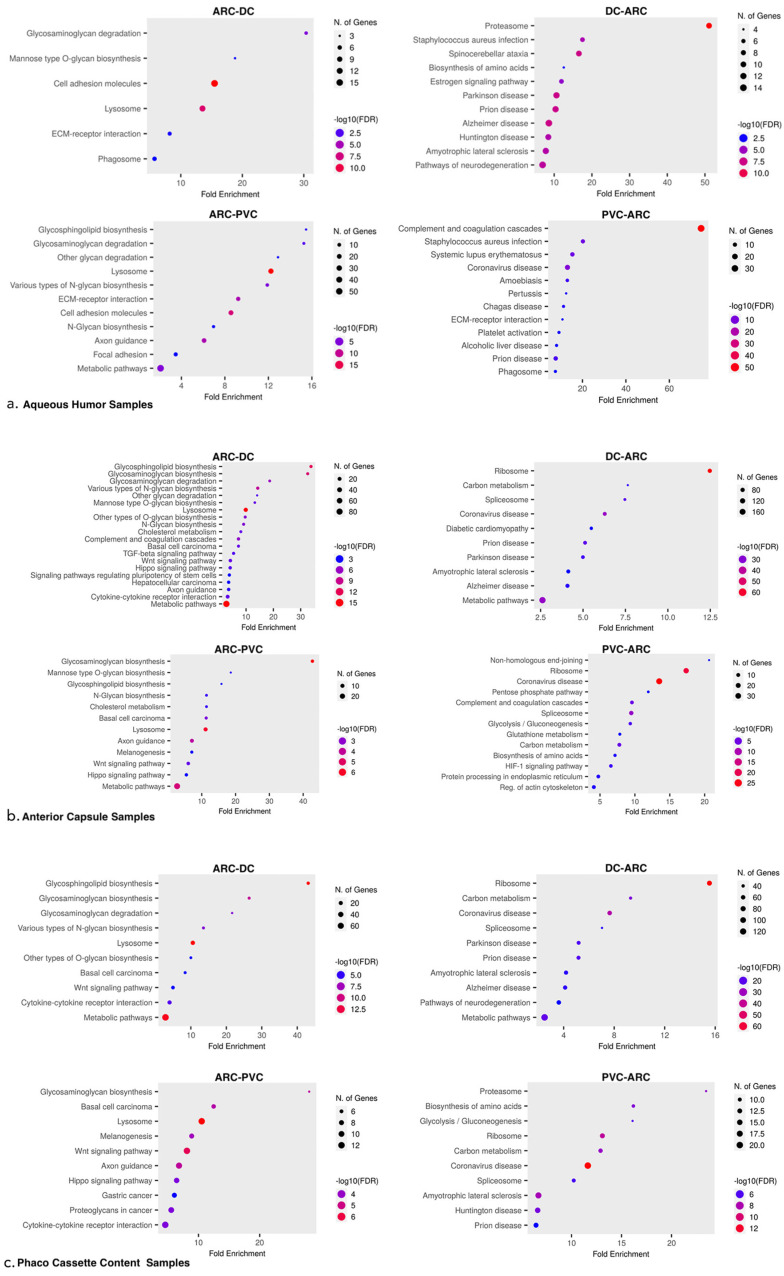
(**a**) Aqueous humor samples. Dotplot of gene enrichment analysis for proteins significantly more abundant in ARC samples when compared with DC samples (top left). Dotplot of gene enrichment analysis for proteins significantly more abundant in DC samples when compared with ARC sample (top right). Dotplot of gene enrichment analysis for proteins significantly more abundant in ARC samples when compared with PVC samples (bottom left). Dotplot of gene enrichment analysis for proteins significantly more abundant in PVC samples when compared with ARC samples (bottom right). (**b**) Anterior capsule samples. Dotplot of gene enrichment analysis for proteins significantly more abundant in ARC samples when compared with DC samples (top left). Dotplot of gene enrichment analysis for proteins significantly more abundant in DC samples when compared with ARC sample (top right). Dotplot of gene enrichment analysis for proteins significantly more abundant in ARC samples when compared with PVC samples (bottom left). Dotplot of gene enrichment analysis for proteins significantly more abundant in PVC samples when compared with ARC samples (bottom right). (**c**) Phaco cassette content samples. Dotplot of gene enrichment analysis for proteins significantly more abundant in ARC samples when compared with DC samples (top left). Dotplot of gene enrichment analysis for proteins significantly more abundant in DC samples when compared with ARC sample (top right). Dotplot of gene enrichment analysis of proteins that presented with significant changes in degradation in ARC samples when compared with PVC samples (bottom left). Dotplot of gene enrichment analysis of proteins that presented with significant changes in degradation in PVC samples when compared with ARC samples (bottom right).

**Figure 4 proteomes-13-00007-f004:**
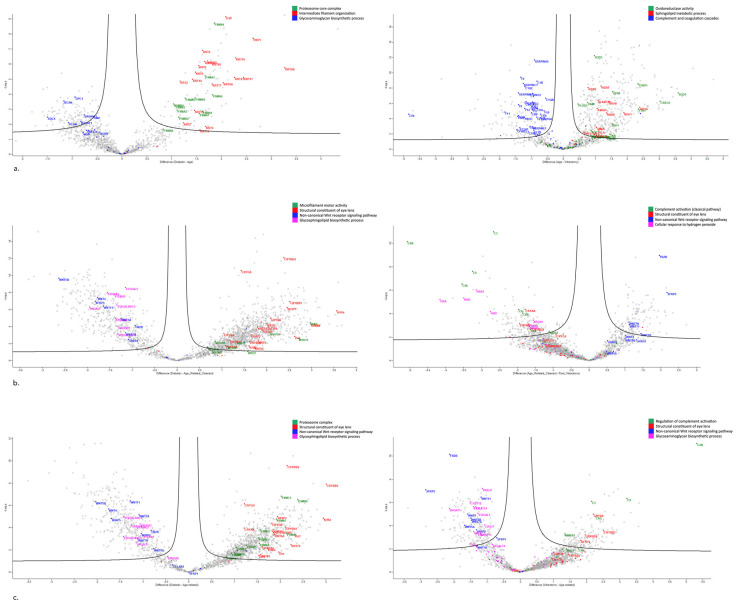
(**a**) Volcano plot of AH sample results, showing DC-ARC (left) and ARC-PVC comparison (right). (**b**) Volcano plot of AC sample results, showing DC-ARC (left) and ARC-PVC comparison (right). (**c**) Volcano plot of phaco cassette content sample results, showing DC-ARC (left) and ARC-PVC comparison (right).

**Figure 5 proteomes-13-00007-f005:**
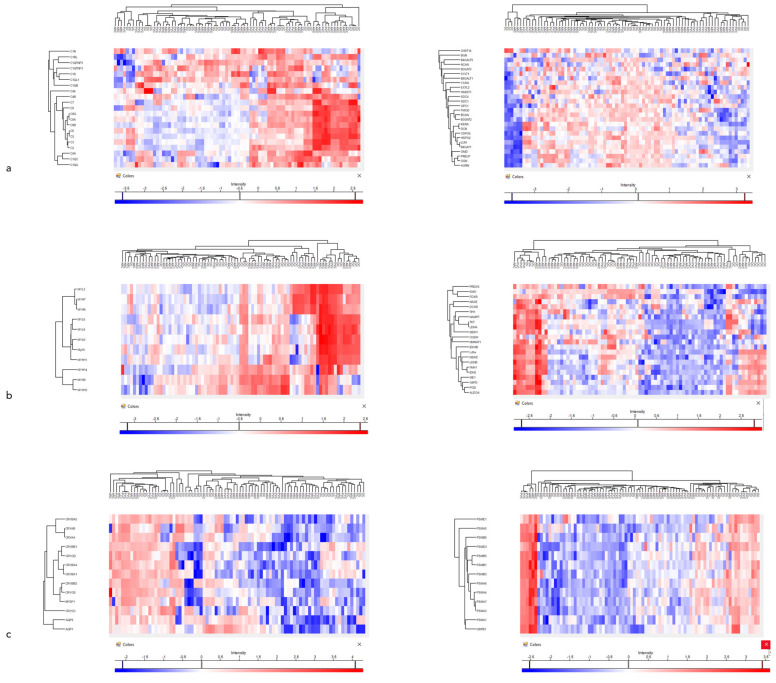
(**a**) Aqueous humor sample heatmaps of proteins involved in complement activation (left) and in glycosaminoglycan biosynthetic process (right). (**b**) Anterior capsule samples heatmaps of myosins (left) and of proteins involved in oxidoreduction coenzyme metabolic process. (**c**) Phaco cassette content sample heatmaps of principal cytoskeletal proteins (left) and of proteins involved in proteasome complex.

**Figure 6 proteomes-13-00007-f006:**
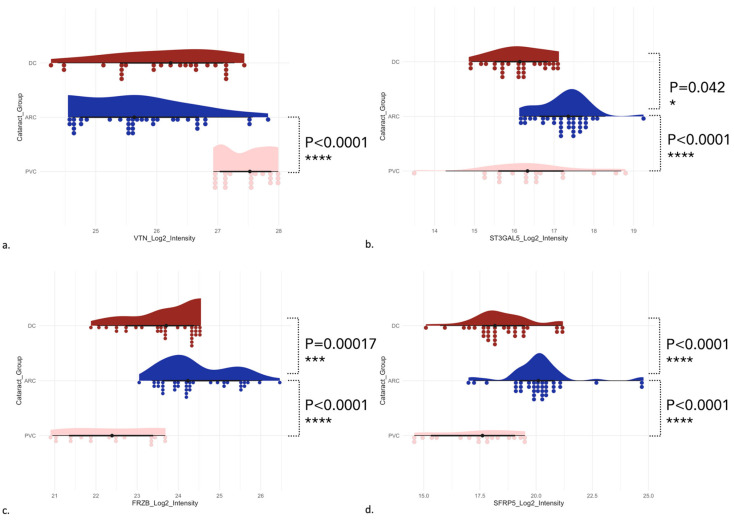
Raincloud plots of VTN protein involved in complement and coagulation cascades in AH samples (**a**) (ARC-DC *p* = 0.193353, ARC-PVC *p* < 0.0001), ST3GAL5 protein involved in glycosphingolipid biosynthetic process in AC samples (**b**) (ARC-DC *p* < 0.0421014, ARC-PVC *p* = 0.0218166), FRZB (**c**) (ARC-DC *p* = 0.00017, ARC-PVC *p* < 0.0001), and SFRP5 proteins (**d**) involved in non-canonical Wnt signaling receptor pathway (ARC-DC *p* < 0.0001, ARC-PVC *p* < 0.0001) in phaco cassette content samples.

**Table 1 proteomes-13-00007-t001:** Cohort composition.

	Group 1 (DC)	Group 2 (ARC)	Group 3 (PVC)
Subjects	11	12	7
Mean age (years, mean ± SD)	61.7 ± 4.3	79.6 ± 4.2	60 ± 10.2
Sex (male/female)	7:4	5:7	2:5
OD:OS	7:4	6:6	1:6
Dominant type of cataract (NS:CS:PSC)	5:1:5	8:2:2	3:0:4
Mean height(cm, mean ± SD)	171.45 ± 7.13	164.08 ± 9.98	168.88 ± 7.08
Mean weight(kg, mean ± SD)	94.55 ± 6.82	70.58 ± 13.75	77.88 ± 13.35
Mean AL(mm, mean ± SD)	23.10 ± 0.80	23.69 ± 0.97	24.52 ± 1.49
Mean K1(D, mean ± SD)	43.02 ± 0.64	42.89 ± 1.44	41.51 ± 1.51
Mean K2(D, mean ± SD)	44.17 ± 0.79	43.58 ± 1.55	42.65 ± 1.61
Mean sun exposure(hours, mean ± SD)	3.5 ± 2.5	2.2 ± 1.5	4.3 ± 2.4
Use of sunglasses (Yes/No)	7:4	6:6	3:4
Iris color(brown/hazel/blue)	8:2:1	9:3:0	5:1:1
Smoking (Yes/No)	4:7	3:9	0:7
Alcohol consumption (Yes/No)	1:10	2:10	0:7
Hypertension (Yes/No)	2:9	7:5	4:3
Glaucoma (Yes/No)	0:11	1:11	0:7
Aspirin intake (Yes/No)	1:10	1:11	0:7
AMD (Yes/No)	0:11	4:8	0:7
Thyroid disease (Yes/No)	2:9	3:9	0:7
Diet supplementary intake (Yes/No)	3:8	5:7	1:6

SD = standard deviation; OD = right eye; OS = left eye; NS = nuclear sclerotic cataract; CS = corticoid spoking cataract; PSC = posterior subcapsular cataract; AL = axial length; K1 = flat meridian of the anterior corneal surface; K2 = steep meridian of the anterior corneal surface; D = Diopters; AMD = age-related macular degeneration.

## Data Availability

The mass spectrometry proteomics data have been deposited to the ProteomeXchange Consortium via the PRIDE partner repository with the dataset identifier PXD045547, PXD045554, PXD045557 [72].

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
