# Peer review of "Differential Signaling Pathways Identified in Aqueous Humor, Anterior Capsule, and Crystalline Lens of Age-Related, Diabetic, and Post-Vitrectomy Cataract"

_proteomes, 2025, doi:10.3390/proteomes13010007_

Round 1

Reviewer 1 Report

Comments and Suggestions for Authors

This paper presents a very interesting study the protein content of three sample types from three cataract types using advanced proteomics technology. Comparisons of proteomes from such samples provides valuable insight into cataractogenesis.  The experiments appear to be well-done; however, I have some concerns with data presentation and interpretation.

Specific points of concern:

1.       It is not ideal to compare results from aqueous humor to phacocassette samples. As the authors acknowledge, these two sample types, as well as anterior capsule samples, represent completely different tissue compartments containing completely different cell types (AC - epithelium and phaco-fiber). The AH sample contains a blood filtrate and few cells and is therefore completely different in terms of the proteins present. Thus, the proteomes are expected to be different in these samples. Further, when comparing cataract groups, it is essential to indicate what samples are being compared and not group all sample types (AC, AH, and phaco) together.

2.       Another issue is the lack of normal lenses as a source of age-matched control sample to distinguish age-related changes from cataractogenic changes that are likely different in each group. While this limitation is addressed in the last section of the manuscript, aged clear lens samples can be obtained from eye banks.

3.       The authors use the terms up-regulation and down-regulation in describing results from phaco samples. My understanding is that protein abundance in fiber cells is altered by modification rather than regulation of protein synthesis or degradation.

4.       The authors should be more specific in describing data in Figures 3 and 4. Rather than describing the pathways as "significantly impacted". The authors should indicate direction of change in specific samples.

Author Response

We wish to express our appreciation to the Editor and Reviewer for their insightful comments, which have helped us significantly to improve our manuscript. According to the suggestions, we have thoroughly revised our manuscript and its final version is enclosed (Any changes in the manuscript are highlighted). Point-by-point responses to the comments are listed below. We remain at your disposal for any further recommendations for revision.

Reviewer 4

This paper presents a very interesting study the protein content of three sample types from three cataract types using advanced proteomics technology. Comparisons of proteomes from such samples provides valuable insight into cataractogenesis.  The experiments appear to be well-done; however, I have some concerns with data presentation and interpretation. 

Specific points of concern: 

  1. It is not ideal to compare results from aqueous humor to phacocassette samples. As the authors acknowledge, these two sample types, as well as anterior capsule samples, represent completely different tissue compartments containing completely different cell types (AC - epithelium and phaco-fiber). The AH sample contains a blood filtrate and few cells and is therefore completely different in terms of the proteins present. Thus, the proteomes are expected to be different in these samples. Further, when comparing cataract groups, it is essential to indicate what samples are being compared and not group all sample types (AC, AH, and phaco) together.

Response: We would like to thank the reviewer for their thoughtful remark. All comparison have been performed between cataract groups of each sample type. No comparisons were made between different samples types. If there was a similar trend in all 3 samples types in a specific cataract group, this was discussed. We remain at your availability, in case you would like any specification.

  1. Another issue is the lack of normal lenses as a source of age-matched control sample to distinguish age-related changes from cataractogenic changes that are likely different in each group. While this limitation is addressed in the last section of the manuscript, aged clear lens samples can be obtained from eye banks. 

Response: We would like to thank the reviewer for their precious comment. Indeed, we are currently doing our best to find control samples from phacoemulsification surgery, but this process may take time. The control group will validate our results, but in the current study, with 3 different cataract groups, each one can serve as control for the other two.

  1. The authors use the terms up-regulation and down-regulation in describing results from phaco samples. My understanding is that protein abundance in fiber cells is altered by modification rather than regulation of protein synthesis or degradation.

Response: We would like to thank the reviewer for pointing this out. We have revised our manuscript accordingly, and used “significantly different” instead of “up- / down-regulation”.

  1. The authors should be more specific in describing data in Figures 3 and 4. Rather than describing the pathways as "significantly impacted". The authors should indicate direction of change in specific samples.

Response: We would like to thank the reviewer for highlighting this. We used the phrase “significantly impacted” because without normal lenses we could not describe up- or down- regulation. We have accordingly revised our manuscript.

Reviewer 2 Report

Comments and Suggestions for Authors

The manuscript by Karakosta et al explores changes in proteomes of the lens, aqueous humor and anterior capsule of human patients with age-related cataract, diabetic cataract and post vitrectomy cataract. The study is valuable as it analyzed human samples and uses good statistical analytical methods. The authors discuss limitations t the end of the manuscript, the main being that no transparent lenses were used in this work. There are a few details that need more description and clarification.

Abstract: Define Sp3 protocol

Line 43  Replace crystalline with ocular 

Line 54  Cantrell et al add year of publication.

Line 116  Deg C deg should be superscript

Remove: Spaces before the word proteome

Change: Arterior to anterior in Fig 1A

What was the volume of the phaco cassette?

How did the authors separate AH from capsule and lens phaco extract. It is not clear from Figure 1. It looks like they were combined in the phaco cassette.

Line 220: How was it determined that the proteins from AH, capsule and lens did not cross contaminate?

Line 455: incomplete sentence.

Line 460 should it be CRYAA?

Author Response

Response to reviewers’ comments

We wish to express our appreciation to the Editor and Reviewer for their insightful comments, which have helped us significantly to improve our manuscript. According to the suggestions, we have thoroughly revised our manuscript and its final version is enclosed (Any changes in the manuscript are highlighted). Point-by-point responses to the comments are listed below. We remain at your disposal for any further recommendations for revision.

Editor

  1. I highlighted some similar sentences in the manuscript, please refer to 
    the attachment to reduce the similarity rate of your manuscript.
    Response: We would like to thank the editor for their time and effort. We have revised the manuscript in all the highlighted sentences to reduce the similarity index.

  2. Please rewrite "Author Contributions" according to Proteomes style and 
    ICMJE guide.

Response: We would like to thank the editor for their kind remark. We have accordingly rewritten "Author Contributions" according to Proteomes style.

Reviewer 1

The manuscript by Karakosta et al explores changes in proteomes of the lens, aqueous humor and anterior capsule of human patients with age-related cataract, diabetic cataract and post vitrectomy cataract. The study is valuable as it analyzed human samples and uses good statistical analytical methods. The authors discuss limitations at the end of the manuscript, the main being that no transparent lenses were used in this work. There are a few details that need more description and clarification.

Response: We would like to thank the reviewer for their kind comment.

Abstract: Define Sp3 protocol

Line 43  Replace crystalline with ocular  

Line 54  Cantrell et al add year of publication.

Line 116  Deg C deg should be superscript

Remove: Spaces before the word proteome

Change: Arterior to anterior in Fig 1A

Line 455: incomplete sentence.

Line 460 should it be CRYAA?

Response: We would like to thank the reviewer for highlighting these points. We have addressed all these points.

What was the volume of the phaco cassette?

Response: We would like to thank the reviewer for pointing this out. The volume of the phaco cassette varied and it nay reached up to 250ml.

How did the authors separate AH from capsule and lens phaco extract. It is not clear from Figure 1. It looks like they were combined in the phaco cassette.

Line 220: How was it determined that the proteins from AH, capsule and lens did not cross contaminate?

Response: We would like to thank the reviewer for their kind remark. In figure 1 each graph represent the steps of the surgery and the sample collection. In the beginning of the surgery all the AH was collected from the anterior chamber. When the capsuolorrhexis was completed, it was washed with BSS and stored in a sterile plastic box with BSS. At the end of the surgery, the phaco cassette was collected, and as already described in the manuscript, since we expected some re-secretion of the AH during these 5 minutes of the surgery, the bag would include the lens fragments and a small portion of the re-secreted AH. However, since between AH and cassette samples there were only 30 proteins in common (Figure 2), we excluded those and could safely assume that the rest came from the lens fragments. We have revised the Figure 1 caption to address this point.

Reviewer 3 Report

Comments and Suggestions for Authors

1- When the authors shows in table 1 the cohort composition, concerning the dominant type of cataracts the diabetic group is very similar to the postvitrectomy patients, but I didn’t see comparisons between them, all the statistics were made against the age related group,  could be possible to see this results ?

2-Concerning methods; this is the first study to use phaco cassette for this issue, the main question is: we don’t  know the cataract stage and hardness of the patients that could affect the results;did the authors register the phaco time and the energy delivered in every surgery?

3-the lack of control group is another weakness of this study, we have been published elsewhere control groups taken from the implant of prevista line lens in mild to moderate young myopes and clear lens extraction from hyperopes or myopes without any kind of associated pathologies, we appreciate this group like a controls, authors could have the opportunity to look  them.

4-There  are many pathways that are affected like Wnt receptors ,Vimentin ,BFSP1 and 2 , the well known aquaporines etc, which  of them have more relevant paper for therapeutic implications?

5-Finally in the discussion section authors speaks about the posterior capsule opacification related to decorin proteoglycans ,¿did the authors find any differences among  the groups that could predict a future capsule opacification ?

Author Response

We wish to express our appreciation to the Editor and Reviewer for their insightful comments, which have helped us significantly to improve our manuscript. According to the suggestions, we have thoroughly revised our manuscript and its final version is enclosed (Any changes in the manuscript are highlighted). Point-by-point responses to the comments are listed below. We remain at your disposal for any further recommendations for revision.

Reviewer 2

1-When the authors shows in table 1 the cohort composition, concerning the dominant type of cataracts the diabetic group is very similar to the postvitrectomy patients, but I didn’t see comparisons between them, all the statistics were made against the age related group,  could be possible to see this results ?

Response: We would like to thank the reviewer for pointing this out. Since the current study did not include a control group of clear lenses, we used the other two groups (DC and ARC) in order to cross check the differences between them and the ARC, serving as controls of each other. Comparisons between DC and ARC were made but they were not discussed in the current study because it did not meet the focus of it and in addition the length of the manuscript would be extensively long and difficult to follow. We are more than happy to share with you the results, if interested.

2-Concerning methods; this is the first study to use phaco cassette for this issue, the main question is: we don’t  know the cataract stage and hardness of the patients that could affect the results;did the authors register the phaco time and the energy delivered in every surgery?

Response: We would like to thank the reviewer for their precious comment. This is the reason why all surgeries where performed by the same experienced surgeon to minimize overall surgery and phaco time. The delivered energy was not recorded but taken into account that the operation time was minimized to 5 minutes, we differences, if any, are neglectable.

3-the lack of control group is another weakness of this study, we have been published elsewhere control groups taken from the implant of prevista line lens in mild to moderate young myopes and clear lens extraction from hyperopes or myopes without any kind of associated pathologies, we appreciate this group like a controls, authors could have the opportunity to look  them.

Response: We would like to thank the reviewer for their thoughtful comment. We are currently collecting control samples of clear lens extraction from hyperopes, since myopia may involve additional pathways for cataract formation. We really hope that the samples will be collected and analyzed soon, so that we can extend the current study and validate our results.  

4-There  are many pathways that are affected like Wnt receptors ,Vimentin ,BFSP1 and 2 , the well known aquaporines etc, which  of them have more relevant paper for therapeutic implications?

Response: We would like to thank the reviewer for highlighting this. Since cataract is developed as a result of dysregulation of several different pathways, we could only highlight every possible therapeutic target suggested by our results, for future research.

5-Finally in the discussion section authors speaks about the posterior capsule opacification related to decorin proteoglycans ,¿did the authors find any differences among  the groups that could predict a future capsule opacification ?

Response: We would like to thank the reviewer for their precious comment. This is a great area for further research which was not investigated in the current study.

Reviewer 4 Report

Comments and Suggestions for Authors

The authors profiled the proteomes from three sample types of cataract patients. The patients were grouped based on their causes of cataracts. This study is novel and interesting. However, the results could be presented more attractive. 

1. To reduce confusion, please change all groups 1-3 to the name of diseases (DC, ARC, PVC), like labels in figures. 

2. The results of pathways could be listed in tables for easy reading and comparison. It will be helpful to rank pathways using the FDR.

3. The discussion was presented as a review article. It could be more attractive to discuss the common and special proteins in three sample types or three causes. 

Author Response

We wish to express our appreciation to the Editor and Reviewer for their insightful comments, which have helped us significantly to improve our manuscript. According to the suggestions, we have thoroughly revised our manuscript and its final version is enclosed (Any changes in the manuscript are highlighted). Point-by-point responses to the comments are listed below. We remain at your disposal for any further recommendations for revision.

Reviewer 3

The authors profiled the proteomes from three sample types of cataract patients. The patients were grouped based on their causes of cataracts. This study is novel and interesting. However, the results could be presented more attractive. 

1.To reduce confusion, please change all groups 1-3 to the name of diseases (DC, ARC, PVC), like labels in figures. 

Response: We would like to thank the reviewer for their precious comment. In the figures and in the text all groups are labelled based on the disease to avoid confusion.

  1. The results of pathways could be listed in tables for easy reading and comparison. It will be helpful to rank pathways using the FDR.

Response: We would like to thank the reviewer for their thoughtful remark. We have included the pathways in an excel file in the supplementary material. For easy reading and comparison, we have included the volcano plots which summarize the principal differentially expressed pathways.

  1. The discussion was presented as a review article. It could be more attractive to discuss the common and special proteins in three sample types or three causes. 

Response: We would like to thank the reviewer for pointing this out. Since the volume of our data was quite large and included differences between both groups and sample times, it would be of no value if not discussed at all. We tried to add a short comment on each different pathway and we used the sub-headings to give readers the opportunity to focus on the area of their interest.